# Diagnostic Delay in Paediatric Inflammatory Bowel Disease—A Systematic Investigation

**DOI:** 10.3390/jcm11144161

**Published:** 2022-07-18

**Authors:** Bahareh Sophia Khalilipour, Andrew S. Day, Kristin Kenrick, Michael Schultz, Kristina Aluzaite

**Affiliations:** 1Gastroenterology Research Unit, Department of Medicine, Dunedin School of Medicine, University of Otago, Dunedin 9016, New Zealand; b.khalilipour@gmail.com (B.S.K.); michael.schultz@otago.ac.nz (M.S.); 2Department of Paediatrics, University of Otago Christchurch, Christchurch 8011, New Zealand; andrew.day@otago.ac.nz; 3Department of General Practice and Rural Health, Dunedin School of Medicine, University of Otago, Dunedin 9016, New Zealand; kristin.kenrick@otago.ac.nz

**Keywords:** diagnostic delay, IBD, Crohn’s disease, ulcerative colitis, paediatric, children

## Abstract

Diagnostic delays (time from the first symptoms to diagnosis) are common in inflammatory bowel disease (IBD) and may lead to worse disease progression and treatment outcomes. This study aimed to determine the duration of diagnostic delays (DD) and to explore associated factors in a cohort of children with IBD in New Zealand. In this study, patients with IBD diagnosed as children and their parents/caregivers completed questionnaires on the patients’ medical history, diagnostic experience, and demographic characteristics. The parent/caregiver questionnaire also included the Barriers to Care Questionnaire (BCQ). Patients’ healthcare data was reviewed to summarise the history of clinical visits and determine symptoms. Total DD, healthcare DD, patient DD and parent DD were derived from the primary dataset. Factors associated with the different types of DD were explored with a series of simple linear and logistical ordinal regressions. A total of 36 patients (Crohn’s disease 25, ulcerative colitis 10; male 17) were included. They were diagnosed at a median age of 12 years (interquartile range (IQR) 10–15 years). Total healthcare delay (from first healthcare visit to formal diagnosis) was median (IQR) 15.4 (6.5–34.2) months. The median (IQR) specialist-associated delay was 4.5 (0–34) days. Higher household income was associated with shorter healthcare delay (*p* < 0.018), while lower overall BCQ scores (indicating more barriers experienced) were associated with longer total healthcare DD. Higher scores in each subscale of BCQ (Skills; Pragmatics; Expectations; Marginalization; Knowledge and Beliefs) were also significantly associated with shorter total healthcare delay (*p* < 0.04). This study found substantial diagnostic delays in paediatric patients with IBD and identified significant associations between longer total healthcare diagnostic delays and overall household income and higher self-reported barriers to accessing healthcare.

## 1. Introduction

Inflammatory Bowel Diseases (IBD) are a group of chronic inflammatory conditions of the gastrointestinal tract. The two main types are Crohn’s disease (CD) and ulcerative colitis (UC) [1,2,3]. The aetiology of IBD is unclear, but they are typically characterized by the onset of a range of gastrointestinal symptoms such as abdominal pain, diarrhoea and haematochezia. Some individuals will present with more atypical symptoms, such as anaemia or extraintestinal manifestations [1,2,3,4,5]. The challenges of these diseases include variable disease course and severity of symptoms in addition to risk of surgery and side effects of medication, which can affect patients’ quality of life [5]. Due to atypical disease presentation such as extra intestinal symptoms, timely diagnosis is not always straightforward.

Individuals diagnosed with IBD may have variable duration of symptoms prior to the diagnosis, also known as diagnostic lag or diagnostic delay (DD). There are three types of DD—patient-associated delay, parent-associated delay and healthcare-associated delay. Potential reasons contributing to longer DD include poor health literacy and long waits due to limited resource availability [6]. Some studies have found that as many as ~20% of children are diagnosed 12 months or more after their symptoms first appear [4,7]. Similarly, other investigations found median delays between 2 months and 4.5 months, [7,8,9,10,11,12,13,14], while two additional reports [15,16] demonstrated delays as medians of over 8 months.

Prolonged DD poses ongoing disease burdens for the patient and potentially results in worse disease and treatment outcomes and lasting physical and psychological damage. Occult or uncontrolled inflammatory activity in children may lead to growth failure while also increasing the risk for the development of complicated disease that may prompt intestinal surgery [15,16]. In addition to physical consequences, prolonged unmanaged IBD symptoms may translate to lost school days, decreased self-esteem and increasing social isolation that may have long-lasting effects [17].

The increasing incidence of IBD worldwide pressures healthcare systems to meet the needs of the increasing number of patients with IBD [18]. Increasing rates of IBD have been noted in children in New Zealand [19,20]. Reducing the DD could be one way of decreasing this burden [3]. No studies have estimated DD in the New Zealand paediatric population, and, to our knowledge, no in-depth studies have been conducted to investigate the barriers to timely IBD diagnosis in children. The aims of this study were to (i) estimate the total DD in children in New Zealand; (ii) determine what proportion of the DD is attributed to patient-associated delay and healthcare-associated delay; and (iii) explore the underlying demographic and socioeconomic factors associated with DD.

## 2. Materials and Methods

### 2.1. Study Sample

Study participants were recruited from a regional Southern District Health Board (SDHB) IBD database “Episoft” (Sydney, Australia) and South Island paediatric IBD patient database [21]. Patients known to have been diagnosed with IBD before their 18th birthday were mailed invitations to participate in the study. The package included study information, consent forms and the study questionnaires in paper (TeleForm, version 16.2, 2017, OpenText, ON, Canada) and a link to an online version (REDCap, version 10.1.12, Vanderbilt University, Nashville, TN, USA) for the patients and their parents/caregivers [22,23]. Prepaid return envelopes were included. Patients who did not respond within a month from the initial contact were mailed a second invitation package and/or were contacted via phone or email.

### 2.2. Study Questionnaries

Study questionnaires were designed by an interdisciplinary research team and included questions on the timing and type of the first IBD symptoms, reasons for delays in notifying healthcare providers, the experience of the diagnostic process and any treatment methods sought.

The parent/caregiver questionnaire also included the Barriers to Care Questionnaire (BCQ) [24] that has been validated in children with chronic illness and special healthcare needs [25]. The BCQ questionnaire consists of five subscales, namely: skills, pragmatics, expectations, marginalisation and knowledge and beliefs (further detailed in the original publication). The BCQ total scale and each of the subscales is scored 0–100, with higher scores being indicative of lower experienced barriers to utilise healthcare [24].

The data from completed paper questionnaires were transferred to a Microsoft Excel spreadsheet using data collection program TeleForm (version 16.2, 2017, OpenText, Waterloo, ON, Canada), and all the entries were manually curated. Responses from paper questionnaires and REDCap were merged in the Excel sheet.

### 2.3. Healthcare Data

All available healthcare data from up to three years before the formal diagnosis date were reviewed for the consented patients (Figure 1). These data included records from the SDHB database (Health Connect South), general practitioners’ clinical databases and the Southern Community Laboratories’ (SCL) database.

One member of the research team (BSK) extracted information on all the potentially relevant clinical visits for each patient in the pre-diagnostic period. Clinical visits were considered potentially relevant if they reported symptoms indicative of IBD. Data extractions were conducted in consultation with two gastroenterologists (AD and MS) and further reviewed by KA for consistency.

### 2.4. Data Extraction

Time points of interest extracted from the healthcare data, patient and parent questionnaires are summarised in Table A1. The data of interest included the patients’ healthcare visits from the first symptoms to formal diagnoses (Figure 2). Formal diagnosis was defined as the endoscopy date. The following variables were derived:**Patient-associated delay**: Time from the first symptoms to notifying parent/caregiver.**Parent-associated delay**: Time from parents noticing the symptoms to seeking healthcare.**Healthcare delay**: From the first clinical visit to formal diagnosis.
a.**Pre-specialist delay**: First healthcare visit to specialist visit/formal diagnosis.b.**Specialist delay**: Time from the first specialist visit to formal diagnosis.**Total DD**: From first symptoms (as reported by the patient/parent) to formal diagnosis of disease.**Number of clinical contacts**: Number of in-person visits to health professional with complaints related to the IBD diagnosis were counted. Additional forms of contact, such as phone calls, emails and others were counted separately. The number of hospitalisations was also counted for each participant.

### 2.5. Statistical Analysis

Descriptive statistics were derived to summarise the demographic features, key disease symptoms, DD and associated barriers’ data. Measures of central tendency were set as mean and median, while variability was defined as standard deviation (SD) and 25th to 75th percentile (IQR), respectively. These measures were derived for the continuous data and proportions for the categorical data. Series of simple linear regressions were performed to model the effects of relevant variables on the total healthcare DD (continuous estimate from healthcare data) (Table A4). Logistic ordinal regression was used to model the patient- and parent-associated DD (categorical responses in the questionnaire). Explanatory variables for the models were derived from the available literature and based on the clinical experience of the research team. Multiple adjustment was not performed due to the exploratory nature of the study and the small sample size. All the analyses were performed using the statistical programming language R [26]. Statistical significance was defined as *p* < 0.05, but findings approaching significance were reported for the sake of completeness.

## 3. Results

### 3.1. Data Summary

A total of 38 (27%) of the 139 invited patients consented to participation in the study (Figure 1). One of the 38 subjects did not complete the study questionnaire, and one was excluded as the patient was diagnosed after their 18th birthday, making a final study number of 36. A total of 9 questionnaires were completed on REDCap and 27 on paper; 28 parents/caregivers of the 36 participating patients with IBD completed the parent/caregiver questionnaire (5 via REDCap and 23 on paper).

### 3.2. Study Sample

The majority of patients (33/35) stated their ethnicity as New Zealand European, while a minority of Asian (1/35), Middle Eastern/Latin American/African (1/35) and Mixed Māori and European (2/33) were also represented. The included patients were diagnosed at a median (IQR) 12 (10–15) years of age with either CD (*n* = 25) or UC (*n* = 10) (Table 1). The median (IQR) time since diagnosis (at the point of study) was 9.5 (4–12) years.

### 3.3. Healthcare-Associated Diagnostic Delays

Data from 28 of the 36 patients were available for healthcare delay analysis. For these patients, the total healthcare delay was a median (IQR) 15.4 (6.5–34.2) months with a full range of 1.4–107 months. Within the healthcare delay, the median (IQR) specialist-associated delay was 4.5 (0–34) days. The specialist delay estimates included four extreme values of 169, 217, 518 and 582 days. Over these periods, the study participants had a median (IQR) of 9 (5–14) healthcare visits prior to seeing a specialist (range 1–41 visits) and a median (IQR) of 1 (1–2) specialist visits before their diagnosis with IBD (range 1–5 visits) (Figure 2).

### 3.4. Patient-Associated Delays

Although the largest proportion of patients (*n* = 8/33) waited one to four weeks to inform someone about their symptoms, some (*n* = 3/33) took more than a year. Most of the patients (*n* = 28/33) reported their initial symptoms to their parent or caregiver, subsequent to which the majority (*n* = 28/33) saw a general practitioner (GP) or family doctor as their first healthcare visit (Table 2).

### 3.5. Parent-Associated Delays

A total of eighteen of the 28 parents surveyed reported having no prior knowledge about IBD and its symptoms (Table A3). Consistent with the patient questionnaire, most parents (*n* = 18/28) reported that they first became aware of their child’s symptoms when their child told them, and 21/28 contacted their GP as the first healthcare contact. The total score in the BCQ was a median (IQR) of 61 (48–74). The specific questionnaire scores are summarised in Table A5.

### 3.6. Healthcare Diagnostic Delays: Simple Linear Regression Analysis 

Patients’ sex (*p* = 0.75) or type of IBD diagnosis (*p* = 0.66) did not predict the length of healthcare DD. Furthermore, the educational level of the parents/caregivers (*p* > 0.24) or their knowledge of IBD (*p* = 0.19) also did not predict the duration of the healthcare DD. However, above average household income predicted shorter healthcare DD when compared to lower than average household income. Those with above average income had a mean (SD) 79 (82) % shorter healthcare DD than those with below average income (*p* = 0.018).

BCQ results were found to be predictive of the healthcare delay (Figure 3 and Table A4). Lower scores (indicating more barriers experienced) in skills (*p* = 0.0016), pragmatics (*p* = 0.031), expectations (*p* = 0.0015), marginalization (*p* = 0.044) and knowledge and beliefs (*p* = 0.0035) and the total BCQ score (*p* = 0.0024) were associated with longer healthcare delays. A 10-point increase in any of the subscales (scale 1–100) resulted in a mean (SD) 16 (9) % to 37 (10) % decrease in healthcare DD (depending on the subscale) (Table A4).

The dataset contained two extreme healthcare delay values of over five years. The analyses were performed with the complete dataset and with exclusion of the two extreme cases. The findings were consistent between the two datasets (Table A4).

### 3.7. Patient and Parent-Associated Delay: Logistic Ordinal Regression Outcomes 

Female patients had a significantly higher risk of increased patient-associated DD (*p* = 0.032) (Table A2). Households with higher education had lower probability of longer patient-associated delay (*p* = 0.030) but not parent-associated delays (*p* = 0.12). Higher household income appeared to be associated with decreased risk of longer patient-associated DD (*p* = 0.06). Analysis did not identify any other predictors of patient- or parent-associated delays. A full summary of analysis outcomes is reported in (Table A2).

## 4. Discussion

This study involved an in-depth investigation of the nature and duration of diagnostic delays in individuals diagnosed with IBD in childhood in New Zealand and found substantial healthcare-, patient- and parent-associated delays. Factors that significantly influenced the healthcare-associated diagnostic delays in this cohort included household income, self-reported healthcare-utilisation knowledge, skills and pragmatics as captured by the BCQ questionnaire.

The length of total DD outlined in the current study exceeds delays from some previously published work, where median total DD was reported to be between 3 months and 4.5 months [7,8,9,10,11,12,13,14,27]. These articles all focused on paediatric patients in Western Europe except two that where from Canada [7] and Israel [11]. However, two other paediatric studies from Saudi Arabia and New Zealand reported substantially longer total DD with medians of two years and 8.4 months [28,29]. In contrast, specialist delays in the current study were considerably shorter than those described in a Spanish paediatric study by Jiménez et al. [10]. The pre-specialist delay of 15.3 months exceeds the findings in another paediatric report from Saudi Arabia of 8 months for CD and 5 months for UC [30]. The majority (75%) of the healthcare visits prior to seeing a diagnosing specialist were visits to a general practitioner (GP).

The total median (IQR) score in the Barriers to Care Questionnaire (BCQ) was 61 (48–74). It was found that a 10-point increase in any of the questionnaire subscales (which translates to fewer reported barriers) was associated with lower healthcare DD (16 (9)% to 37 (10)% depending on the specific subscale). While there are multiple factors implicated, this finding highlights an important set of systemic and patient-level barriers that result in worse clinical outcomes. Self-reported barriers such as pragmatics, feelings of marginalisation and health system knowledge and beliefs were all significantly associated with diagnostic delay outcomes. A recent study by Stamm et al. (2020) [31] found a low gastroenterologist count per population and gaps in gastroenterology specialist coverage across New Zealand. While Stamm et al. (2020) [31] did not focus on paediatric gastroenterology services, it captured an important systemic challenge associated with wide geographical dispersion of the population of New Zealand that could be expected to contribute to diagnostic delays. Similarly, Basu et al. (2019) [32] found referral delay in cancer patients associated with economic and geographical barriers to healthcare. It is important to highlight that a relatively small increase in BCQ score resulted in a substantial decrease in diagnostic delays, which may suggest some important targets for future interventions to decrease diagnostic delays in this patient population.

There are many reasons for DD, but accessible information, a user-friendly and intuitive healthcare system with ensured continuity of clinical care and easier access to healthcare could help decrease healthcare diagnostic delays [32,33]. Further studies are required to identify the most effective interventions to decrease these barriers.

The current study found that children from households with higher than average incomes had a shorter healthcare DD compared to those with lower than average incomes. This could be due to the patients from lower than average households facing more logistical obstacles, such as difficult access to healthcare or not having the same financial freedom to take time off for doctors’ visits. However, a recent Austrian study found no influence of household income on total DD [3]. It is important to highlight that direct comparisons between countries are complicated by differences in healthcare systems, economic and geographic factors, among others.

In the current study, 42% of the subjects waited less than one month before seeking help, while the same was true for 61% of the parents/caregivers seeking healthcare support for their children. These results are shorter than the patient-associated delay presented in Nahon et al. (2014) [30], who found a median (IQR) delay of 6 (1–24) months for CD and 2.4 (0.3–7) months for UC. In contrast to this, the patient-associated delay found in the current study is slightly longer than those reported by Treviño et al. (2020) [10], which found an overall median (IQR) of 13.8 (6.9–32.4) days.

A likely contributor to the length of DD could be the type and severity of the symptoms, with more atypical symptoms prolonging the time it takes to suspect a potential IBD diagnosis. For instance, patients with haematochezia had a shorter time to diagnosis than those without haematochezia in one report [34]. The current study also gathered detailed information on the type and character of symptoms at disease onset, both from medical records and from patients and their parents/caregivers. The most common symptoms reported by patients were abdominal pain, diarrhoea, tirednes, and blood in the stool. Patients who presented with atypical symptoms, such as muscle pain, joint pain or skin rash, had longer delays that included 8 to 33 healthcare visits. However, a detailed analysis on the type of symptoms and patient-/parent-associated DD was not conducted due to the small sample size.

The majority of the parents/caregivers became aware of their child’s symptoms when the child informed them. With younger children, parents/caregivers rely more on non-verbal communication explaining a possible association between younger age and longer healthcare DD. In contrast, older children are more independent and self-aware, which can delay the parents/caregivers’ awareness due to the nature of the symptoms. This trend of longer patient delay was also significant in female patients. Some patients specifically reported this in their responses to the questionnaires. An association between younger age and longer DD has been noted in some previous reports [35,36]. However, one report found no relationship between age at diagnosis and DD [37], while several reports have indicated longer DD in older patients [38,39,40].

The limitations to the study were first, potential recall bias in questionnaire responses since the included patients were diagnosed a median of 9.5 years prior to the study. However, the healthcare DD data were derived from clinical records, and should not be impacted by the time since diagnosis. Second, there is potential selection bias, as patients who had strong positive or negative experiences during their pre-diagnostic period could be more inclined to participate. Third, while the BCQ was validated in children with asthma [24,25], it has not been validated in patients with IBD. The strengths of the study include a detailed investigation incorporating clinical records and responses from, patients and parents/caregivers.

In conclusion, the current study identified long total and healthcare DD in children diagnosed with IBD in NZ and highlighted potential factors that impede timely diagnosis, including prolonged periods of DD before seeing a specialist. Future research is required on how to effectively address the barriers to timely diagnostic delay. Possible interventions could include enhancing modes of communication between GPs and specialists as well as fast track systems to enable prompt assessment of patients with suspected IBD [41].

## Figures and Tables

**Figure 1 jcm-11-04161-f001:**
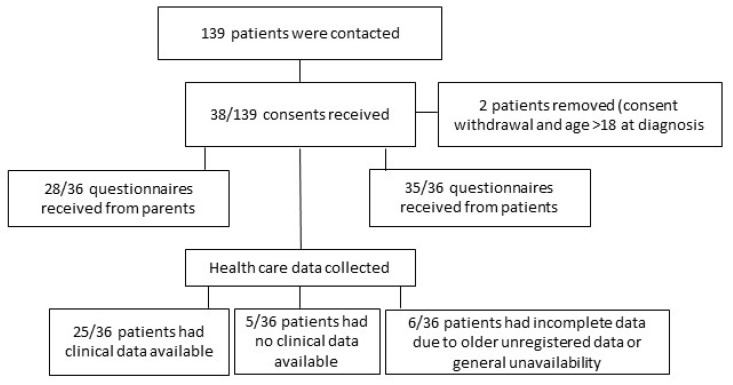
Study participation and the responses received.

**Figure 2 jcm-11-04161-f002:**
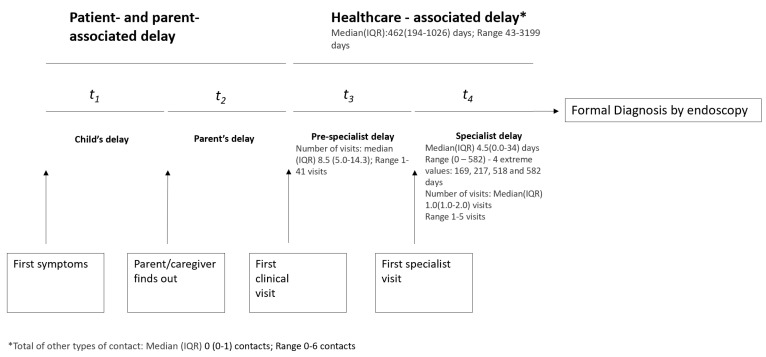
Diagnostic delay timeline and the variables of interest.

**Figure 3 jcm-11-04161-f003:**
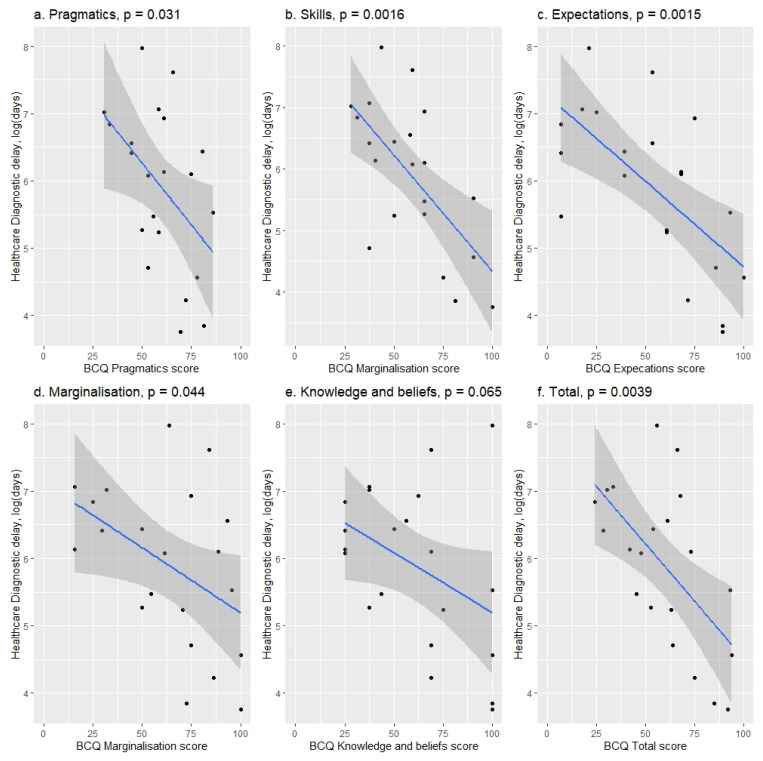
BCQ subscales relation to total healthcare delay. Higher scores on each individual subscale in the BCQ questionnaire correspond to lower healthcare delay. Median (IQR) for each subscale: pragmatics, 61 (51–72); skills, 59 (44–70); expectations, 61 (38–73); marginalisation, 66 (48–85); knowledge and beliefs, 63 (41–72).

**Table 1 jcm-11-04161-t001:** Demographic characteristics of study participants (patients with inflammatory bowel disease and their parents/caregivers).

Patient Characteristics	% (*n*)
**Age at the time of the study**, years	
Median (IQR)	22 (15–28)
Range	8–72
**Time since diagnosis** *, years	
Median (IQR)	9.5 (4–12)
Range	0–59
**Age at diagnosis**, years	
Median (IQR)	12 (10–15)
Range	0–17 years
**Sex**, male	47 (17/36)
**Ethnicity**	
NZ European **	94 (33/35)
Asian	3 (1/35)
MELAA (Middle Eastern/Latin American/African)	3 (1/35)
Mixed Māori and European	6 (2/33)
**Diagnosis**	
Crohn’s disease	71 (25/35)
Ulcerative Colitis	29 (10/35)
**Parent/caregiver characteristics**	% (*n*)
**Ethnicity**	
NZ European	89 (25/28)
Other European	7 (2/28)
MELAA (Middle Eastern/Latin American/African)	4 (1/28)
**Highest level of education in the household (at the time of diagnosis)**	
High school or less	18 (5/28)
Apprenticeship or advanced	54 (15/28)
University degree	29 (8/28)
**Employment (at the time of diagnosis)**	
Both employed	3 (10/28)
One employed	25 (7/28)
None employed	0 (0/28)
At least one employed (only info from one parent)	39 (11/28)
**Income *** (at the time of diagnosis)**	
Below average	21 (6/28)
Middle	54 (15/28)
Above average	25 (7/28)

Some missing data for one patient, hence the different denominators for Ethnicity and Diagnosis summary statistics. * In time since diagnosis, only 2 patients were diagnosed 40 and 59 years ago. ** Out of whom 1/33 were of mixed Asian ethnicity. *** Average income at the time of the study in New Zealand was NZD 100.103 according to: https://www.stats.govt.nz/information-releases/household-income-and-housing-cost-statistics-year-ended-june-2017 (accessed on 1 February 2020).

**Table 2 jcm-11-04161-t002:** Patient questionnaire: IBD symptoms and reporting prior to diagnosis (*n* = 35).

Question	% (*n*)
**What were your initial symptoms?** (*multiple choices possible*)	
Abdominal pain	74 (26/35)
Diarrhoea	69 (24/35)
Tiredness	63 (22/35)
Blood in stool	60 (21/35)
Weight loss	54 (19/35)
Nausea	26 (9/35)
Vomiting	23 (8/35)
Pus in stool	11 (4/35)
Constipation	9 (3/35)
Fever	6 (2/35)
**How long after you first noticed symptoms did you tell someone?**	
Less than a week	18 (6/33)
1 week to less than 1 month	24 (8/33)
1 month to less than 2 months	15 (5/33)
2 months to less than 1 years	15 (5/33)
More than 1 year	9 (3/33)
Do not know/ remember	18 (6/33)
**Who did you tell first about the symptoms?**	
Parent/caregiver	85 (28/33)
GP/family doctor	6 (2/33)
Other—parents/caregivers noticed/the patient was an infant	9 (3/33)
**Which healthcare professional did you see first?**	
GP/family doctor	85 (28/33)
Multiple	3 (1/33)
Emergency Department	3 (1/33)
Do not know/remember	9 (3/33)
**Overall experience of the diagnostic process:**	
Very good	12 (4/33)
Good	9 (3/33)
Neither good nor bad	30 (10/33)
Bad	24 (8/33)
Very bad	9 (3/33)
Do not know/ remember	15 (5/33)

Full questionnaire responses were not available for 2 patients.

## Data Availability

De-identified raw data will be made available upon request.

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
