# Peer review of "Diagnostic Delay in Paediatric Inflammatory Bowel Disease—A Systematic Investigation"

_jcm, 2022, doi:10.3390/jcm11144161_

Round 1

Reviewer 1 Report

The authors presented an in-depth study of the causes of diagnostic delay in children with IBD in New Zealand through questionnaire surveys from the patients, and parents/ caretakers, and clinical database system for healthcare providers. It is a well-designed study, despite a small case number.

Below are several points to be added for the completeness of this manuscript:

1.      Please add some background epidemiology data (incidence and prevalence) and trends (increasing?) of childhood IBD in New Zealand in the Introduction.

2.      From Table 1, the time range since diagnosis is quite variable (0-59 years), indicating the diagnostic periods also spanned an extended time range. Although the case number was small, did the author find any cohort effect (period effect) along the length of time? Did the trend parallel the epidemiological trend, especially in the HCP-related DD?

3.      As the authors stated, DD may have an impact on patient outcomes. Please describe the effect (such as bowel resection rate in 1 year or 5 years) on the patients with variable DD.

4.      In section 3.7, the authors disclosed that female patients had a higher risk of patient-related DD by logistic regression. It is against common sense that females are more likely to be health seekers. What is the gender difference in the incidence/prevalence of childhood IBD in New Zealand? In this study, more females patients participated in the questionnaire survey. Is it possibly biased?

5.      In the Discussion, the authors stated that the low workforces of gastroenterologists in New Zealand are an important factor in HCP-related DD. How about the DD in adult IBD patients in New Zealand? Is there a significant difference between the DD in childhood and adult IBD? Besides, Ref 30 lacks the Journal identity.

Author Response

28 June 2022

To the editors,

Thank you for facilitating the review of our paper “Diagnostic Delay in Paediatric Inflammatory Bowel Disease – a systematic investigation” and considering the resubmission of the improved version. We have addressed each of the points raised by the reviewers in detail below.

[Reviewer 1]

The authors presented an in-depth study of the causes of diagnostic delay in children with IBD in New Zealand through questionnaire surveys from the patients, and parents/ caretakers, and clinical database system for healthcare providers. It is a well-designed study, despite a small case number. Below are several points to be added for the completeness of this manuscript:

  1. Please add some background epidemiology data (incidence and prevalence) and trends (increasing?) of childhood IBD in New Zealand in the Introduction.

We thank the reviewer for the positive feedback on our manuscript. We have added the following sentence with two additional  references to the Introduction (lines 60-61): “Increasing rates of IBD have been noted in children in New Zealand1,2.”

  1. Lopez RN, Evans HM, Appleton L, Bishop J, Chin S, Mouat S, et al. Prospective Incidence of Paediatric Inflammatory Bowel Disease in New Zealand in 2015: Results From the Paediatric Inflammatory Bowel Disease in New Zealand (PINZ) Study. J Pediatr Gastroenterol Nutr. 2018 May;66(5):e122-e126
  2. Lopez RN, Appleton L, Gearry RB, Day AS. Rising Incidence of Paediatric Inflammatory Bowel Disease in Canterbury, New Zealand, 1996-2015. J Pediatr Gastroenterol Nutr. 2018 Feb;66(2):e45-e50

  1. From Table 1, the time range since diagnosis is quite variable (0-59 years), indicating the diagnostic periods also spanned an extended time range. Although the case number was small, did the author find any cohort effect (period effect) along the length of time? Did the trend parallel the epidemiological trend, especially in the HCP-related DD?

Thanks to the reviewer for the excellent comment. We agree that this type of analysis would be very interesting and valuable. However, we did not have a sufficient number of participants to conduct such analysis. Given this comment, we realise that the range of years-since-diagnosis presented in the Table 1 is misleading. We had 2 patients with 40 and 59 years since the diagnosis and several others with more than 15 years since diagnosis. For most of these patients little to no data were available, and hence they were excluded from some of the analyses. We have added a note next to the table 1 to highlight that only 2 patients had their diagnosis of IBD 40 and 59 years ago (lines 160).

With the remainder of the patients, we did not think we had sufficient sample size to explore the period effect, but we agree this could be of interest for future studies.

  1. As the authors stated, DD may have an impact on patient outcomes. Please describe the effect (such as bowel resection rate in 1 year or 5 years) on the patients with variable DD.

We agree that this would be an important aspect to consider. The current study design did not permit longitudinal assessment of outcome or progress. We feel that this question may be best answered with a more population-based cohort or perhaps in a prospective fashion.

  1. In section 3.7, the authors disclosed that female patients had a higher risk of patient-related DD by logistic regression. It is against common sense that females are more likely to be health seekers. What is the gender difference in the incidence/prevalence of childhood IBD in New Zealand? In this study, more females patients participated in the questionnaire survey. Is it possibly biased?

At present, the reasons for this are  unclear. Epidemiological data from NZ indicates that more boys than girls are diagnosed in childhood. There are no NZ data focusing on rates of DD in adults. The reason for more women answering than men does provide a bias (and the reason for this is unclear), although in the current study the difference in numbers among subjects is small with 17 men and 19 women.

  1. In the Discussion, the authors stated that the low workforces of gastroenterologists in New Zealand are an important factor in HCP-related DD. How about the DD in adult IBD patients in New Zealand? Is there a significant difference between the DD in childhood and adult IBD? Besides, Ref 30 lacks the Journal identity.

To our knowledge there are no data of DD in adults in NZ.

Thank you for your comment about Reference #30: we have corrected the details provided for this reference (line 448).

[Reviewer 2]

Thank you for the possibility to review the manuscript entitled “  Diagnostic Delay in Paediatric Inflammatory Bowel Disease – a systematic investigation”  The subject matter discussed in the article is current and very important, indicating prolonged periods of DD before seeing a specialist as a main factor that impede timely diagnosis. The article needs to be slightly improved before it can be accepted for publications.

  1. The small study group is the major limitation of the work. What is the likely reason for such a weak response?

As outlined in the discussion, the exact reason for this is unclear. The opt-in study design that was required to include initial contact by post and additional follow-up, is likely to be the primary reason.

  1. Table 2. In my opinion questionnaire of IBD patients should include the question about extraintestinal symptoms. These symptoms may precede gastrointestinal symptoms.

The questionnaire used asked the patients and their parents on presentation of a list of symptoms, including extraintestinal symptoms (without labelling them as such) [Table 2]. We have also included an in-depth review of any mention of extraintestinal symptoms in the clinical notes for each patient [Table A1].

  1. 2.2. Study questionnaire line 80: note that there are symptoms from gastrointestinal tract. These symptoms are included later in the healthcare professional questionnaire.

We used questionnaires to collect data from patients and their parents to investigate any initial symptoms, behaviours and barriers (section 2.2). We then reviewed available clinical data to investigate the patients’ path through the healthcare system to obtain the diagnosis (we did not use questionnaires for healthcare professionals) (sections 2.3 and 2.4).

  1. Discussion line 219: Please describe in which populations total DD was described. It would be easier for the readers

I will add country, confirm the study is paediatric in brackets after first mentioning them.  

We hope that we adequately addressed the concerns raised by the reviewers, and that the manuscript is now suitable for publication in the journal of Journal of Clinical Medicine. We are very happy to address any further concerns and/or questions.

Yours sincerely,

Authors

Reviewer 2 Report

Thank you for the possibility to review the manuscript entitled “  Diagnostic Delay in Paediatric Inflammatory Bowel Disease – a systematic investigation”  The subject matter discussed in the article is current and very important, indicating prolonged periods of DD before seeing a specialist as a main factor that impede timely diagnosis. The article needs to be slightly improved before it can be accepted for  publications. 

Comments to the Author:

1.     The small study group is the major limitation of the work. What is the likely reason for such a weak response?

2.     Table 2. In my opinion questionnaire of IBD patients should include  the question about extraintestinal symptoms. These symptoms may precede gastrointestinal symptoms.  

3.     2.2. Study questionnaire line 80: note that there are symptoms from gastrointestinal tract. These symptoms are included later in the healthcare professional  questionnaire.

4.     Discussion line 219: Please describe in which populations total DD was described. It would be easier for the readers

Author Response

(The authors gave the same response as above.)
